# Integrated metabolomics and bioactivity analysis of new chrysanthemum cultivar petals: Insights into eye-protecting agents

Yifan Wang[1], Hedi Zhao[1], Gangqiang Dong[2]*, Dan Yang[1], Xiaofei Liu[2], Zhuyin Chen[1], Yanyan Su[2], Yan Liu[1], Jingjing Zhu[1], Jun Zhang[1], Jing Zhang[1]*, Sha Chen[1]*

1 State Key Laboratory for Quality Assurance and Sustainable Use of Dao-Di Herbs, Institute of Chinese Materia Medica, China Academy of Chinese Medical Sciences, Beijing, China, 2 Amway (China) Botanical R&D Centre, Wuxi, China

* tony.dong@amway.com (GD); jingzhang1116@163.com (JZ); schen@icmm.ac.cn (SC)

## Abstract

Chrysanthemum is a globally significant economical crop. Through broadly targeted metabolomics, 58 polyphenols and 65 xanthophylls were quantitatively and qualitatively assessed in chrysanthemum petals. Carotenoids including β-cryptoxanthin, lutein stearate, lutein distearate, lutein dioleate, and lutein oleate, emerged as the primary compounds in both ligulate and tubular flowers, and their levels remaining relatively high and stable during the second to fifth stages of chrysanthemum flower development. However, the compound content varied across developmental stages and tissues. Additionally, the activity of a novel chrysanthemum cultivar extract was evaluated; at a concentration of 62.5 μg/mL, it exhibited significantly greater anti-apoptotic effects than glutathione ($p < 0.001$). Moreover, various concentrations of the chrysanthemum extract demonstrated a clear trend in preventing retinopathy. This study integrated chemical composition analysis with activity evaluation to offer fresh insights into the mechanism underlying the development of eye-protecting agents derived from chrysanthemum flowers.

## 1. Introduction

*Chrysanthemum morifolium* Ramat. is a crucial crop utilized globally both as food source and a medicinal agent, which has significant economic value. The flowering head of chrysanthemum is extensively employed as dietary supplement and consumed in the form of herbal tea. The flower contains significant quantities of flavonoids and phenolic components, with corresponding biologically active compounds commonly documented [1,2]. Pharmacological investigations have revealed that chrysanthemum species harbor secondary compounds exhibiting diverse biological activities, as along with properties such as brightening or eye-protection [3].

**Data availability statement:** All relevant data are within the manuscript and its Supporting Information files.

**Funding:** This study received funding from New Variety Development Project of Amway (China) Botanical R&D Centre (Grant No. BC20220016Z, awarded to SC). The funder had no role in study design, data collection and analysis, decision to publish, or preparation of the manuscript.

**Competing interests:** The authors have declared that no competing interests exist.

**Abbreviations:** APCI, atmospheric pressure chemical ionization; MRM, multiple reaction monitoring; MS, mass spectrometry; MTC, maximum tolerated concentration; QC, quality control; ESI, electrospray ionization; GSH, glutathione; MS, mass spectrometry; DMSO, dimethyl sulfoxide; BHT, 2,6-di-tert-butyl-p-cresol; LC, liquid chromatography; UPLC, ultraperformance LC; dpf, day postfertilization; OPLS-DA, orthogonal partial least squares-discriminant analysis; $\log_2$FC, $\log_2$ fold change.

Nevertheless, there have been few reports on the combined chemical and pharmacological activities of chrysanthemum.

Chrysanthemum extract has demonstrated various beneficial effects such as anti-inflammatory, anti-bacterial, anti-oxidant and anti-hypertensive properties, attributed in part to its high contents of flavonoids. Additionally, its ability to enhance eye-brightness may be attributed to carotenoids [4–6]. Carotenoids, encompassing carotene and xanthophyll, are $C_{40}$ isoprenoids possessing a terpenoid moiety, crucial for human health as antioxidants and precursors to vitamin A. Typically responsible for petal colors ranging from yellow-to-orange, carotenoids play pivotal roles in chrysanthemum petals. In a previous study, 13 carotenoid compounds were identified phytochemical separation methods [4]. Moreover, utilizing electrospray ionization (ESI), researchers detected 4 carotenoids and 13 luteolin derivatives in chrysanthemum [7]. Consequently, there is an urgent need for a methodology to detect abundant polyphenols and carotenoids in chrysanthemum. The polarity distribution range of carotenoids is extensive, and atmospheric pressure chemical ionization (APCI) offers a means to convert both carotene and oxygenated carotenoids into various positively/negatively charged molecules. Protonated and deprotonate ions are effectively employed in carotenoid analysis [8]. Investigating the detailed carotenoid composition of new chrysanthemum petals is therefore of utmost importance.

Carotenoids in humans are exclusively acquired through dietary intake [9]. Among the 50 reported carotenoids identified in the human diet, they are regarded as significant bioactive compounds. The traditional benefits associated with improved eyesight are also attributed to carotenoid and lutein constituent. Since animals (with the exception of certain aphid species) lack the ability to synthesize carotenoids, they must rely on dietary sources. Consequently, there exists a commercial demand for carotenoids across the food, pharmacy, and cosmetic sectors [4]. The utilization of plant-derived carotenoids has emerged as a prevailing trend, with a pressing need for novel varieties rich in lutein components. Chrysanthemum cultivars are recognized as promising reservoirs of carotenoids for commercial extraction. Furthermore, the transformation system of chrysanthemum is highly developed, necessitating research into new cultivars with enhanced commercial viability.

Assessing novel plant food sources is a complex endeavor that demands careful deliberation. Moreover, evaluating new plant varieties is integral to their commercial utilization in food and herbal tea production. Our study undertook a comprehensive analysis encompassing chemical composition and biological activities. The hypothesis posited that the new chrysanthemum variety may harbor abundant carotenoids, potentially resulting in enhanced bioactive properties. This methodology promises to yield fresh insights into the mechanism underlying the development of eye-protective foods derived from chrysanthemum flowers.

## 2. Materials and methods

### 2.1. Reagents and chemicals

Reduced L-glutathione (GSH, Catalog No. SLCG8572) and Augentropfen Stulln® Mono (eye drops containing esculin and digitalis glycosides, Catalog No. 22J026) were

procured from Pharma Stulln (Stulln, Germany). Acetone (mass spectrometry [MS] grade) was acquired from Merck & Co. (New Rahway, NJ, USA), while analytical-grade dimethyl sulfoxide (DMSO, Catalog No. BCCD8942) and MS-grade formic acid were sourced from Sigma-Aldrich (St. Louis, MO, USA). Acetonitrile, methanol, metal tertiary butyl ether, n-hexane, and 2,6-di-tert-butyl-p-cresol (BHT) of MS grade were obtained from Merck. The carotenoid standards zeaxanthin, β-carotene, lycopene, and lutein were purchased from Shanghai Yuanye Bio-technology Co., Ltd. (Shanghai, China). Chlorogenic acid (Catalog No. L-007–160504, > 98%), galuteolin (Catalog No. RFS-M025019090, purity > 98%), luteolin 7-O-glucuronide (Catalog No. M-041–150603, purity > 98%), 3,4-O-dicaffeoylquinic acid (Catalog No. RFS-Y06911812012, purity > 98%), apigenin-7-O-glucoside (Catalog No. RFS-Q04401902012, purity >98%), lutein (Catalog No. M-007–150916, purity >98%), apigenin (Catalog Q-002-140731-1, purity >98%), diosmetin (Catalog No. CHB190217, purity > 98%), 3,5-O-dicaffeoylquinic (Catalog No. CHB151013, purity >98%), and 4,5-O-di caffeoylquinic acid (Catalog No. CHB151013, purity >98%) were purchased from the RunFenSi Technology Limited (Beijing, China). Methylcellulose (Catalog No. C2004046), mycophenolate mofetil (Catalog No. H14J6335), acridine orange (Catalog No. C15109250), and cobalt chloride (Catalog No. J2228429) were obtained from Shanghai Aladdin Biochemical Technology Co., Ltd. (Shanghai, Beijing).

## 2.2. Plant materials and sample extraction

The new chrysanthemum cultivar and the control cultivar were cultivated at the WuXi Amway Plant Research and Development Center (Wuxi, China, 31°57′ N, 120°30′ E). Samples were systematically collected from six distinct growth stages, designated as S1 Fig, between late October to early November (Fig 1A). The initial stage, S1, corresponded to the bud stage, followed by early bloom stages (S2 and S3) at 2-day intervals. The blooming stage (S4) was reached next, followed by the late-blooming stage (S5) after 3 days. Finally, the decay stage (S6) was observed 21 days later. Chrysanthemum samples at these growth stages were collected over one-month period. All flower heads were subjected to enzyme deactivation by heating at 100°C for 2 minutes and subsequently dried to a constant weight at 45°C for 14 hours. Equal amounts of dried samples were used for carotenoid extraction.

   **2.2.1. Carotenoid extraction.** A total of 100 mg of the ground sample were weighed and extracted with 1 mL of n-hexane/acetone/ethanol (1:1:1, v/v/v) solution containing 0.01% BHT (g/mL). The mixture was gently swirled for 20 min at room temperature and subsequently centrifuged for 5 min at 12000 rpm at 4°C. The supernatant was extracted twice and then subjected to centrifugation. Finally, both supernatants were combined. The resulting extracted solution was concentrated, re-dissolved in a mixture of 200 μL of methanol/methyl tert-butyl ether (1:1, v/v), filtered through 0.22-μm filters, and transferred to a brown sample vial for liquid chromatography (LC)-MS/MS analysis.

   **2.2.2. Flavonoid extraction.** All dried samples were precisely weighed at 0.10 g, and then extracted by adding 10 mL of a methanol and water solution (70:30, v/v). The mixture underwent ultrasonication for 40 minutes, followed by cooling and re-weighing to ascertain any lost volume. Any lost volume was replenished using 70% methanol, and the mixture was thoroughly shaken well. The resulting supernatants were collected and subsequently filtered through a 0.22-nm Millipore filter (Alltech Scientific, Shanghai, China) before undergoing LC-MS analysis.

   **2.2.3. Chrysanthemum extraction.** The chrysanthemum extract was desiccated to obtain a powder form, then reconstituted in standard diluted water to create a mother liquor solution with a concentration of 2.00 mg/mL.

## 2.3. Preparation of the standard solution

Appropriate quantities of 12 flavonoid and polyphenol standards, along with 4 carotenoid standards were weighed and dissolved in methanol to prepare stock solutions with a concentration of 500 mg/L. These stock solutions were then combined in methanol to generate quality control (QC) samples, which were scheduled for analysis after every ten experimental samples to assess the stability of the LC-MS system. The mixed stock solution of the standards was subsequently diluted by factors of 5-, 20-, 100, or 400, or 1000, and the series concentrations of the solutions were used to create quantitative standard curves.

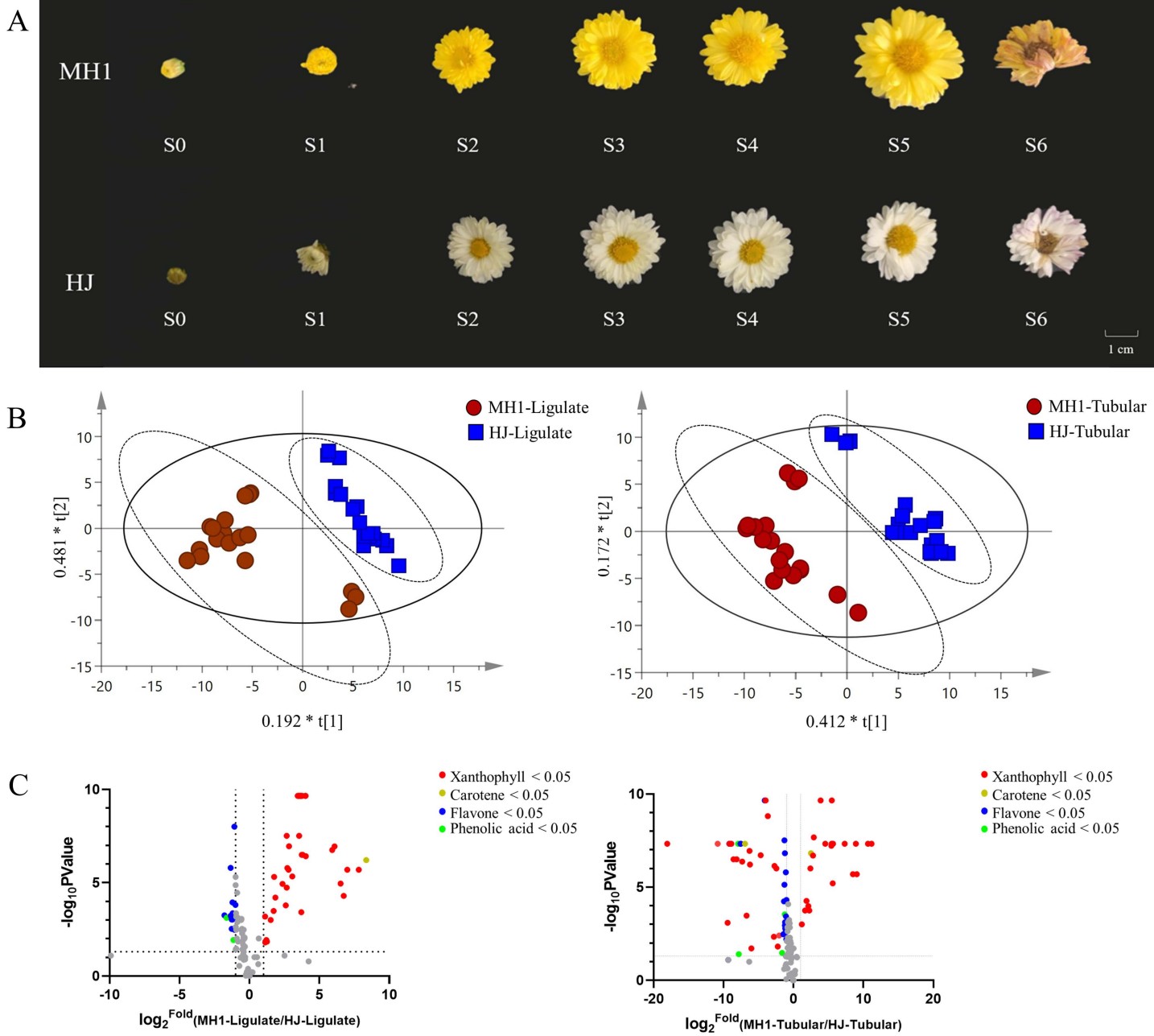

**Fig 1. Relationship between secondary metabolites (carotenoids and flavonoids) and their morphological variations among six different flower growth stages. (A)** Six different flower growth stages of the new cultivars (abbreviated as MH1) and the check variety cultivar (abbreviated as HJ); **(B)** OPLS-DA of MH1 and HJ chrysanthemum flower with secondary metabolites; **(C)** Volcano plots of secondary metabolites.

## 2.4. LC-MS analysis of carotenoids and flavonoids

The quantitative analysis employed ultra-performance liquid chromatography (UPLC)-coupled with tandem triple quadrupole mass spectrometry (MS). Carotenoids were separated using a Nano $C_{30}$ (3 μm, 100 mm × 2.1 mm) column in the UPLC system. The mobile phase consisted of methanol/acetonitrile (1:3, v/v) with 0.01% butylated hydroxytoluene (BHT)

and 0.1% formic acid A, while mobile phase B comprised a methyl tert-butyl ether solution with 0.015 BHT. Gradient elution was performed according to the following scheme: 0–3 min, 100% A; 3–5 min, 100%–30% A; 5–9 min, 30%–5% A; 9–10 min, 5%–100% A; and 10–11 min, 100% A. The flow rate was maintained at 0.8 mL/min, with a column temperature of 25°C, and a sample injection volume was 2 µL.

APCI-MS detection was conducted employing multiple reaction monitoring (MRM) in the negative mode, with the following parameters: a temperature of 350°C and a gas pressure of 25 psi. Quantitative measurements of all carotenoids were executed using optimized de-clustering potential and collision energy settings. In each cycle, spanning a time interval of five minutes, the scanning of ion pairs corresponding to target compounds is sequentially performed to achieve dynamic multiple reaction monitoring (MRM) for quantitative analysis.

A Waters Xevo G2-S-Q-TOF mass spectrometer (Waters, Milford, MA, USA) was equipped with a dual electrospray ionization (ESI) source capable of operation in both positive and negative modes. The ion source temperature for the MS system was maintained at 120°C, and nitrogen (N2) served as the atomized gas. The scan range for parent ion masses was set from 100–1200 m/z. Operational parameters included a voltage of 40 V, a capillary voltage of 2 kV, the collision energy ranging from 20 to 50 eV. Additionally, the impact energy of the parent ion was set at 6 eV, the de-solvent gas flow rate at 600 L/h, and the gas temperature at 400°C. Data acquisition and processing were performed using the MassLynx 4.1 MS workstation.

An Agilent 6460 mass spectrometer (Agilent, Santa Clara, CA, USA) was equipped with a dual ESI electrospray ionization (ESI) source. Mass detection was conducted in the dynamic multiple reaction monitoring (MRM) mode, specifically in the negative polarity, utilizing the following parameters: ion spray voltage of 5500 V; collision energy of 20 eV, ion source gas at 50 psi; collision energy gas at 50 psi; and curtain gas at 30 psi. High performance liquid chromatography (HPLC) separation was performed using an ACQUITY UPLC BEH $C_{18}$ column (1.7 µm, 50 mm × 2.1 mm, Waters, USA). The mobile phases consisted of acetonitrile (mobile phase A) and 0.1% formic acid in water (mobile phase B). Gradient elution conditions for the separation were programmed as follows: 0–8 min, 5%–16% A; 8–12 min, 16%–20% A; 12–15 min, 20%–30% A; 15–18 min, 30%–40%; 18–21 min, 40%–90% A; 21–25 min, 90% A; and 25.1–30 min, 5% A.

## 2.5. Experimental animals

Zebrafish (SYXK2022−0004) were raised reared in fish culture water maintained at a temperature of 28°C. The water quality parameters were as follows: con 200 mg of instant sea salt was added to every 1 L of reverse osmosis water, the conductivity was 450–550 µS/cm, pH was 6.5–8.5, and hardness was 50–100 mg/L $CaCO_3$), which was provided by the company's fish breeding center. The feeding management met the requirements of the AAALAC international accreditation (certification number: 001458) based on the National Institutes of Health guide for the care and use of Laboratory animals. The IACUC ethics review number was IACUC-2023-6963-01.

Transgenic fluorescent zebrafish (Fli-1 strain) at 1 day postfertilization (dpf) were used to determine the maximum tolerated concentration (MTC) of the chrysanthemum extract and evaluate its efficacy in preventing retinopathy.

The wild-type zebrafish AB line was used at 1 dpf to evaluate the prevention of apoptosis in ocular cells treated with the chrysanthemum extract.

The zebrafish facility and the laboratory at Hunter Biotechnology, Inc., are accredited by the Association for Assessment and Accreditation of Laboratory Animal Care International and by the China National Accreditation Service for Conformity Assessment. After experiments, all the zebrafish were anesthetized and euthanized with 0.25 g/L tricaine methanesulfonate, which conforms to the American Veterinary Medical Association requirements for euthanasia by anesthetic. This study was approved by the Institutional Animal Care and Use Committee (IACUC) at Hunter Biotechnology, Inc., and the IACUC approval number was 001458.

## 2.6. MTC determination

Transgenic fluorescent zebrafish (Fli-1 strain) were randomly distributed into six-well plates with 30 zebrafish per well, constituting the experimental group. The chrysanthemum extract was administered in an aqueous solution, with each well

having a capacity of 3 mL. Concurrently, control and model groups were established, with each well having a capacity of 3 mL. Concurrently, control and model groups were established. Excluding the control group, zebrafish in the model group were administered cobalt chloride to induce wet macular degeneration. After treatment at 28°C for 3 days, the maximum tolerated concentration (MTC) of the chrysanthemum extract in the model zebrafish was determined.

## 2.7. Antiapoptotic effects in ocular cells

Wild-type AB zebrafish at 1 day post- (dpf) were randomly distributed into six-well plates, with 30 fish per well. The chrysanthemum extract was administered as an aqueous solution, while 625 µg/mL GSH served as the positive control. Each well had a capacity of 3 mL. Excluding the control group, zebrafish were treated with mycophenolate mofetil in water solution to induce a zebrafish ocular cell apoptosis model. After 1 day of treatment at 28°C, 10 zebrafish were randomly selected from each group and photographed under a fluorescence microscope. The fluorescence intensity of ocular apoptotic cells was statistically analyzed to evaluate the effect of the chrysanthemum extract on apoptosis.

## 2.8. Efficacy in preventing retinopathy

Transgenic fluorescent zebrafish (Fli-1 strain) at 1 day post-fertilization (dpf) were randomly distributed into six-well plates with 30 zebrafish per well. They were then administered the chrysanthemum extract in aqueous solution. The positive control group received 20 µL/mL Augentropfen Stulln® Mono, while the control and model groups were prepared with a capacity of 3 mL per well. Excluding the normal control group, zebrafish were treated with water-soluble cobalt chloride to induce a wet macular degeneration model. After 3 days of treatment at 28°C, 10 zebrafish were randomly selected from each group and photographed using a fluorescence microscope. The ocular choroidal vascular area of zebrafish was measured to evaluate the efficacy of chrysanthemum extract in preventing retinopathy.

## 2.9. Statistical analysis

An orthogonal partial least squares-discriminant analysis (OPLS-DA) score scatter plot was generated using SIMCA 13.0 for interpretation, employing internal 7-fold cross-validation. The horizontal component represents the variation among the groups, while the vertical dimension captures the variation within the groups of the OPLS-DA. Additional, a volcano plot was employed to visualize the different in compounds among the groups. On the x-axis, the $\log_2$ fold change ($\log_2$FC) is depicted, with metabolites exhibiting the greatest differences positioned at the two ends. The y-axis illustrates $-\log_{10}$ P, which denotes the negative logarithm of the P-value of the statistical test. Significance testing was conducted at the $p \leq 0.05$ level utilizing the Student–Newman–Keuls significance test, which was also utilized to assess variation among the development stages. Correlation coefficients of the secondary metabolites were calculated using SPSS 26.0 software (IBM, Armonk, NY, USA). Zebrafish images were analyzed using NIS-Elements D 3.20 software, and data are presented as means ± SE. Statistical analysis was performed using SPSS 26.0 software.

## 2.10. RNA-seq analysis

Total RNA was extracted from frozen tissue using a plant RNA extraction kit. mRNA libraries for each sample were constructed and sequenced on the Illumina NovaSeq 6000 platform. After filtering the raw data, the sequencing error rate and GC content distribution were assessed. Clean reads were subsequently obtained for further analysis, and the mapped data were aligned to the cultivated chrysanthemum reference genome (https://figshare.com/articles/dataset/A_chromosome-scale_genome_assembly_of_hexaploid_cultivated_chrysanthemum/21655364/2). Transcripts per kilobase of transcript per million fragments mapped (TPM) values were used as an indicator of transcript or gene expression levels. Genes with a TPM value greater value than 1 in at least one tissue across all tissues considered were selected for analysis. Weighted Gene Co-expression Network Analysis (WGCNA) was conducted based on (this selection criterion using

the WGCNA software package. All transcriptome datasets generated in this study have been deposited in the NGDC GSA database (PRJCA034797).

## 3. Results

### 3.1. UPLC-MS/MS–based metabolomics analysis of phenolic acids, flavonoids, carotenes, and xanthophylls in the chrysanthemum varieties MH1 and HJ

A total of 55 flavonoid and 68 xanthophyll compounds identified in both the flowers of the new cultivar (abbreviated as MH1) and the reference variety cultivar (abbreviated as HJ) (Tables 1 and 2). These compounds were quantitatively analyzed in the Multiple Reaction Monitoring (MRM) mode. MRM transitions for the compounds were determined by analyzing their fragmentation profiles and structures. To optimize MS conditions, MRM transitions of all compounds were examined in both the positive and negative ion modes. It was observed that carotenes and xanthophylls exhibited higher ion response intensity in the positive mode, while phenolic acids and flavonoids showed higher relative ion response intensity in the negative mode. Consequently, the positive ion mode was employed to detect carotenes and xanthophylls, whereas the negative ion mode was utilized for phenolic acids and flavonoids detection. Collision energy of 123 components was optimized. UPLC conditions were also optimized to achieve satisfactory chromatographic settings, ensuring optimal chromatographic separation and peak symmetry. All 123 identified compounds in the two cultivars (MH1 and HJ) are depicted in the Orthogonal Partial Least Squares- Discriminant Analysis (OPLS-DA) plot.

Technical and analytical errors must be kept sufficiently small to prevent disruptions in multivariate data analysis, thereby ensuring that acquisition of reliable and high-quality data with UPLC-MS/MS–based metabonomics. Line plots of the quality control (QC) samples were generated and plotted between −2 standard deviations (SD) and +2SD.

#### 3.1.1. Differences between MH1 and HJ.
As a new variety, MH1 exhibited yellow coloration in all six stages, while HJ displayed white coloration across the same six stages (Fig 1A). Representative compounds from MH1 and HJ with either tubular or ligulate flowers were identified using OPLS-DA. High predictability ($Q_2$) and strong goodness of fit ($R_2X$, $R_2Y$) were observed in the OPLS-DA models when comparing the ligulate flowers ($Q_2 = 0.192$, $R_2X = 0.481$, $R_2Y = 0.673$) and tubular flowers of MH1 and HJ ($Q_2 = 0.584$, $R_2X = 0.412$, $R_2Y = 0.172$). In the OPLS-DA plot, greater horizontal distance indicated greater differences between the groups, while closer vertical distance indicated better within-group reproducibility. As depicted in Fig 1B, samples from the two cultivars (MH1 and HJ) exhibited significant variability. In summary, the ligulate and tubular flowers of MH1 and HJ displayed clear separation (Fig 1B).

The primary compounds responsible for the variances between the cultivars were 31 xanthophylls and carotenes, identified in both tubular and ligulate chrysanthemum flowers using criteria of $p < 0.05$ and variable importance for projection > 1. The screening results are depicted as volcano plots, as shown in Fig 1C. Gray dots denote nondifferential compounds between the MH1 and HJ cultivars. Negative values indicate compounds with lower levels in MH1 than in HJ (on the right dots), while positive values indicate compounds with higher levels in MH1 compared to HJ (on the left side). In terms of ligulate flowers, the levels of xanthophylls and carotenes, particularly violaxanthin myristate and capsanthin, were higher in MH1 than in HJ ($log_2FC > 10$). Concerning tubular flowers, the levels of lutein dioleate and zeaxanthin dipalmitate were significantly higher in MH1 than in HJ ($log_2FC > 10$), whereas those of β-citraurin and zeaxanthin-oleate-palmitate were lower in MH1 than in HJ ($log_2FC < 10$). Xanthophylls and carotenes as the main differential components between the two cultivars. Phytofluene was the sole carotene exhibiting significantly different levels between the two cultivars.

### 3.2. Qualitative and quantitative analysis of phenolic acids, flavonoids, carotenes, and xanthophylls at different ripening stages

We conducted an analysis of carotenes, xanthophylls, phenolic acids, and flavonoids in the tubular and ligulate flower of MH1 and HJ cultivars across six different ripening stages. As depicted in Fig 2A, the xanthophyll content in ligulate flowers

**Table 1. Computed high-resolution mass, molecular weight, and fragment ions of the phenolic acids, flavonoids detected in UPLC-Q-TOF-MS.**

| No. | $t_R^b$ | Q1 (Da) | Q3 (Da) | Error (ppm) | Molecular formula | Molecular Mass | Identification |
|---|---|---|---|---|---|---|---|
| Polyphenols (parent ions [M-H]⁻ for compounds) | | | | | | | |
| Caffeoylquinic acids | | | | | | | |
| 1 | 1.13 | 353.0858 | 191.0858 | −1.40 | $C_{16}H_{18}O_9$ | 354.0858 | Neochlorogenic acid |
| 2 | 1.68 | 353.0861 | 191.0861 | 1.10 | $C_{16}H_{18}O_9$ | 354.0861 | Chlorogenic acid |
| 3 | 1.87 | 353.0858 | 191,0858 | −1.10 | $C_{16}H_{18}O_9$ | 354.0858 | Cryptochlorogenic acid |
| 4 | 2.29 | 179.0323 | 135.0323 | −0.60 | $C_9H_8O_4$ | 180.0323 | Caffeic acid |
| 12 | 5.55 | 515.1187 | 353.1187 | −1.40 | $C_{25}H_{24}O_{12}$ | 516.1187 | 3,4-O-Dicaffeoylquinic acid |
| 13 | 5.63 | 515.1195 | 353.1195 | 0.20 | $C_{25}H_{24}O_{12}$ | 516.1195 | 1,5-O-Dicaffeoylquinic acid |
| 14 | 5.71 | 515.1201 | 353.1201 | 0.00 | $C_{25}H_{24}O_{12}$ | 516.1201 | 3,5-O-Dicaffeoylquinic acid |
| 22 | 6.47 | 515.1217 | 353.1217 | −1.40 | $C_{25}H_{24}O_{12}$ | 516.1217 | 4,5-O-Dicaffeoylquinic acid |
| 27 | 7.85 | 515.1187 | 191.1187 | 1.60 | $C_{25}H_{24}O_{12}$ | 516.1187 | 1,3-O-Dicaffeoylquinic acid |
| 34 | 8.62 | 515.1180 | 191.1180 | 0.40 | $C_{25}H_{24}O_{12}$ | 516.1180 | 1,4-O-Dicaffeoylquinic acid |
| 39 | 9.33 | 677.1515 | 353.1515 | 0.70 | $C_{34}H_{30}O_{15}$ | 678.1515 | 3,4,5-O-Tricaffeoylquinic acid |
| Flavonoids | | | | | | | |
| 5 | 2.66 | 431.1907 | 269.1907 | 0.20 | $C_{21}H_{20}O_{10}$ | 432.1907 | Apigenin-7-O-β-D-galactoside |
| 6 | 3.66 | 461.1650 | 285.1650 | −1.30 | $C_{21}H_{18}O_{12}$ | 462.1650 | Luteolin-7-glucuronide |
| 7 | 4.00 | 323.0748 | 179.0748 | 2.10 | $C_{15}H_{16}O_8$ | 324.0748 | Skimmin |
| 8 | 4.50 | 595.1672 | 287.1672 | −0.50 | $C_{27}H_{32}O_{15}$ | 596.1672 | Eriodictyol-7-O-β-D-rutinoside |
| 9 | 4.65 | 593.1516 | 285.1516 | 1.00 | $C_{27}H_{30}O_{15}$ | 594.1516 | Luteolin-7-O-β-D-rutinoside |
| 10 | 4.73 | 449.1077 | 287.1077 | −1.30 | $C_{21}H_{22}O_{11}$ | 450.1077 | Eriodictyol-7-O-β-D-glucoside |
| 11 | 4.87 | 447.0927 | 285.0927 | 0.40 | $C_{21}H_{20}O_{11}$ | 448.0927 | Luteolin-7-O-β-D-glucoside |
| 15 | 6.06 | 431.0980 | 269.0980 | −1.40 | $C_{21}H_{20}O_{10}$ | 432.0980 | Apigenin-7-O-β-D-glucoside |
| 16 | 6.16 | 445.0760 | 269.0760 | −0.90 | $C_{21}H_{18}O_{11}$ | 446.0760 | Apigenin-7-O-β-D-glucuronide |
| 17 | 6.23 | 607.1665 | 299.1665 | 0.00 | $C_{28}H_{32}O_{15}$ | 608.1665 | Neodiosmin |
| 18 | 6.25 | 491.1183 | 287.1183 | 0.00 | $C_{23}H_{24}O_{12}$ | 492.1183 | Eriodictyol-7-O-β-D-(6″-O-acetyl)-glucoside |
| 21 | 6.40 | 489.1035 | 285.1035 | 0.60 | $C_{23}H_{22}O_{12}$ | 490.1035 | Luteolin-7-O-β-D-(6″-O-acetyl)-glucoside |
| 25 | 7.28 | 269.0436 | 269.0436 | −1.10 | $C_{15}H_{10}O_5$ | 270.0436 | Genistein |
| 29 | 8.03 | 547.1084 | 299.1084 | −1.30 | $C_{25}H_{24}O_{14}$ | 548.1240 | Diosmetin-7-O-β-D-(6″-O-malonyl)-glucoside |
| 30 | 8.27 | 505.1328 | 301.1328 | −1.00 | $C_{23}H_{22}O_{13}$ | 506.1328 | Quercetin-7-O-β-D-(6″-O-acetyl)-glucoside |
| 31 | 8.32 | 503.1913 | 285.1913 | −1.40 | $C_{24}H_{25}O_{12}$ | 504.1913 | Luteolin-4′-methoxy-7-O-β-D-(6″-O-acetyl)-glucoside |
| 32 | 8.47 | 285.0387 | 285.0387 | −0.70 | $C_{15}H_{10}O_6$ | 286.0387 | Luteolin |
| 33 | 8.59 | 609.2904 | 301.2904 | 1.50 | $C_{27}H_{30}O_{16}$ | 610.2904 | Quercetin-7-O-β-D-rutinoside |
| 35 | 8.68 | 487.1812 | 283.1812 | −1.30 | $C_{24}H_{24}O_{11}$ | 488.1812 | Acacetin-7-O-β-D-(3″-O-acetyl)-glucoside |
| 37 | 9.11 | 445.1124 | 283.1124 | 1.40 | $C_{22}H_{22}O_{10}$ | 446.1124 | Acacetin-7-O-β-D-galactoside |
| 38 | 9.21 | 473.1074 | 269.1074 | −2.30 | $C_{23}H_{22}O_{11}$ | 474.1074 | Apigenin-7-O-β-D-(6″-O-acetyl)-glucoside |
| 41 | 9.66 | 459.0922 | 283.0922 | −1.10 | $C_{22}H_{20}O_{11}$ | 460.0922 | Acacetin-7-O-β-D-glucuronide |
| 42 | 9.99 | 271.0588 | 271.0588 | 0.40 | $C_{15}H_{12}O_5$ | 272.0588 | Naringenin |
| 43 | 10.15 | 269.0434 | 269.0434 | 0.00 | $C_{15}H_{10}O_5$ | 270.0434 | Apigenin |
| 44 | 10.47 | 299.0535 | 299.0535 | −1.70 | $C_{16}H_{12}O_6$ | 300.0535 | Chrysoeriol |
| 45 | 10.61 | 283.0587 | 283.0587 | 0.70 | $C_{16}H_{12}O_5$ | 284.0587 | Acacetin |
| 47 | 11.70 | 287.2205 | 287.2205 | −2.10 | $C_{15}H_{12}O_6$ | 288.2205 | Eriodictyol |
| 48 | 12.36 | 343.0810 | 343.0810 | −0.30 | $C_{18}H_{16}O_7$ | 344.081 | Eupatilin |
| 49 | 12.46 | 487.2537 | 283.2537 | 0.80 | $C_{24}H_{24}O_{11}$ | 488.2537 | Acacetin-7-O-β-D-(6″-O-acetyl)-glucoside |
| 50 | 12.56 | 373.0914 | 373.0914 | −1.30 | $C_{19}H_{17}O_8$ | 374.0914 | 3,5-Dihydroxy-4′,6,7,8-tetramethoxyflavone |
| 51 | 12.76 | 299.1301 | 299.1301 | −1.70 | $C_{16}H_{12}O_6$ | 300.1301 | Diosmetin |

*(Continued)*

**Table 1.** (Continued)

| No. | $t_R{}^b$ | Q1 (Da) | Q3 (Da) | Error (ppm) | Molecular formula | Molecular Mass | Identification |
|---|---|---|---|---|---|---|---|
| Flavonoids (parent ions [M+H]+for compounds) | | | | | | | |
| 19 | 6.29 | 537.1244 | 289.1244 | 0.20 | $C_{24}H_{24}O_{14}$ | 536.1244 | Eriodictyol-7-O-β-D-(6″-O-malonyl)-glucoside |
| 20 | 6.38 | 535.1122 | 287.1122 | 0.40 | $C_{24}H_{22}O_{14}$ | 534.1122 | Luteolin-7-O-β-D-(6″-O-malonyl)-glucoside |
| 23 | 6.62 | 463.1232 | 301.1232 | 1.10 | $C_{22}H_{22}O_{11}$ | 462.1232 | Diosmetin-7-O-β-D-glucoside |
| 24 | 7.25 | 519.1144 | 271.1144 | −2.10 | $C_{24}H_{22}O_{13}$ | 518.1144 | Apigenin-7-O-β-D-(6″-O-malonyl)-galactoside |
| 26 | 7.61 | 519.1193 | 271.1193 | 1.50 | $C_{24}H_{22}O_{13}$ | 518.1193 | Apigenin-7-O-β-D-(6″-O-malonyl)-glucoside |
| 28 | 7.99 | 549.1240 | 301.1240 | −0.70 | $C_{25}H_{24}O_{14}$ | 548.1240 | Diosmetin-7-O-β-D-(6″-O-malonyl)-galactoside |
| 36 | 8.79 | 593.1880 | 285.1880 | 0.80 | $C_{28}H_{32}O_{14}$ | 592.1880 | Acacetin-7-O-β-D-rutinoside |
| 40 | 9.52 | 447.1286 | 285.1286 | −1.10 | $C_{22}H_{22}O_{10}$ | 446.1286 | Acacetin-7-O-β-D-glucoside |
| 46 | 10.62 | 533.1299 | 285.1299 | 1.50 | $C_{25}H_{24}O_{13}$ | 532.1299 | Acacetin-7-O-(6″-malonyl)-glucoside |
| 52 | 13.24 | 359.1129 | 359.1129 | −1.70 | $C_{19}H_{18}O_7$ | 358.1129 | 5-Hydroxy-3',4',6,7-tetramethoxyflavone |
| 53 | 13.76 | 389.1233 | 389.1233 | −0.50 | $C_{20}H_{20}O_8$ | 388.1233 | Artemetin |
| 54 | 15.63 | 282.186 | 282.1860 | 1.40 | $C_{18}H_{35}NO$ | 281.1860 | Oleamide |
| 55 | 16.97 | 593.2276 | 285.2276 | 2.90 | $C_{28}H_{32}O_{14}$ | 592.2276 | Linarin |

was significantly higher in MH1 than in HJ, while the flavonoid content in ligulate flowers was significantly lower in MH1 than in HJ across all six stages.Detailed data are available in Supporting Information S1 Table.

Fig 2B displays the levels of carotenes, xanthophylls, phenolic acids, and flavonoids in tubular flowers of MH1 and HJ during the six stages. Futhermore, the xanthophyll content in tubular flowers was significantly higher in MH1 than in HJ, whereas the flavonoid content was significantly lower in MH1 than in HJ.

The average contents of the five most abundant xanthophylls are illustrated in Fig 2C. Levels of lutein myristate, lutein palmitate, lutein dimyristate, lutein dipalmitate, and lutein in ligulate flowers were significantly higher in MH1 than in HJ (Fig 2C). Simultaneously, the contents in the tubular flowers of MH1 were higher than those in HJ. The most abundant flavonoids, including luteolin-7-O-(6″-malonylglucoside), apigenin-7-O-(6″-malonylglucoside), and apigenin-7-O-β-D-rutinoside, in both ligulate and tubular flowers were relatively higher in HJ than in MH1. Conversely, the average contents of acacetin-7-O-β-rutinoside and diosmetin-7-O-(6″-malonylglucoside) were relatively higher in MH1 than in HJ (Fig 2D).

**3.2.1. The five most abundant lutein and flavonoid compounds at different ripening stages.** As depicted in Fig 3A, the levels of the five most abundant xanthophyll compounds in ligulate flowers were significantly higher in MH1 than in HJ across all six stages. Among them, the contents of lutein myristate, lutein palmitate, lutein dimyristate, and lutein dipalmitate were significantly higher in MH1 than in in HJ during the fifth stage. Among the five most abundant xanthophyll compounds in tubular flowers (Fig 3B), lutein dilaurate and lutein were significantly more abundant in MH1 than in HJ across all six stages.

Fig 3C illustrates that among the five most abundant flavonoids in ligulate flowers, the levels of luteolin-7-O-(6″-malonylglucoside), apigenin-7-O-(6″-malonylglucoside), and apigenin-7-O-β-D-rutinoside levels were lower in MH1 than in HJ across all six stages. In tubular flowers, luteolin-7-O-(6″-malonylglucoside) and apigenin-7-O-(6″-malonylglucoside) levels were lower in MH1 than in HJ across all six stages. Moreover, acacetin-7-O-β-D-rutinoside and apigenin-7-O-β-D-rutinoside levels were lower in MH1 than in HJ across all stages except the first stage.

**3.2.2. Gene expression patterns and metabolite correlations along the MEP pathway in chrysanthemum varieties MH1 and HJ.** Genes with expression values higher than 1 were selected for Weighted Gene Co-expression Network Analysis (WGCNA). A total of 85,912 genes were classified into 39 modules based on their expression patterns (Fig 4a). The correlation between the gene expression matrix of different modules and carotenoid detection results was analyzed, with the correlation coefficients and corresponding e-value displayed at the intersection of modules and

**Table 2. The identified and quantitative carotenes, and xanthophylls in flowers of chrysanthemum.**

| No.[a] | $t_R$[b] | Q1(Da) | Q3 (Da) | Molecular formula | Molecular Mass | Identification |
|---|---|---|---|---|---|---|
| Carotenes (parent ions [M+H]+ for compounds) | | | | | | |
| 05 | 1.92 | 543.5 | 81.2 | $C_{40}H_{62}$ | 542.4852 | phytofluene |
| 06 | 4.99 | 545.3 | 81 | $C_{40}H_{64}$ | 544.5008 | (E/Z)-phytoene |
| 07 | 5.54 | 537.6 | 123.2 | $C_{40}H_{56}$ | 536.4382 | ε-carotene |
| 01 | 5.93 | 537.5 | 123.2 | $C_{40}H_{56}$ | 536.4382 | α-carotene |
| 04 | 6.28 | 537.6 | 177.1 | $C_{40}H_{56}$ | 536.4382 | β-carotene |
| 03 | 7.36 | 537.4 | 177.3 | $C_{40}H_{56}$ | 536.4382 | γ-carotene |
| 02 | 8.25 | 537.4 | 81 | $C_{40}H_{56}$ | 536.4382 | lycopene |
| Xanthophylls (parent ions [M+H]+ for compounds) | | | | | | |
| 57 | 1.58 | 601.4 | 221 | $C_{40}H_{56}O_4$ | 600.4179 | violaxanthin |
| 58 | 1.93 | 601.4 | 565.5 | $C_{40}H_{56}O_4$ | 600.4179 | neoxanthin |
| 68 | 2.79 | 433.3 | 341.1 | $C_{30}H_{40}O_2$ | 432.6 | β-citraurin |
| 55 | 2.86 | 585.5 | 175.4 | $C_{40}H_{56}O_3$ | 584.4229 | antheraxanthin |
| 61 | 3.33 | 597.3 | 147.1 | $C_{40}H_{52}O_4$ | 596.84 | astaxanthin |
| 59 | 4.05 | 551.5 | 175.4 | $C_{40}H_{56}O_2$ | 568.428 | lutein |
| 65 | 4.35 | 601.4 | 109 | $C_{40}H_{56}O_4$ | 600.42 | capsorubin |
| 63 | 4.47 | 585.5 | 109.1 | $C_{40}H_{56}O_3$ | 584.871 | capsanthin |
| 62 | 4.53 | 417.3 | 325.3 | $C_{30}H_{40}O$ | 416.638 | 8'-apo-beta-carotenal |
| 56 | 4.63 | 569.4 | 477.5 | $C_{40}H_{56}O_2$ | 568.428 | zeaxanthin |
| 66 | 4.75 | 565.5 | 203.3 | $C_{40}H_{52}O_2$ | 564.8 | canthaxanthin |
| 64 | 5.09 | 553.5 | 123.1 | $C_{40}H_{56}O$ | 552.43 | α-cryptoxanthin |
| 60 | 5.53 | 553.5 | 177.4 | $C_{40}H_{56}O$ | 552.4331 | β-cryptoxanthin |
| 67 | 5.55 | 551.6 | 203.1 | $C_{40}H_{54}O$ | 550.9 | echinenone |
| 28 | 5.70 | 783.7 | 583.4 | $C_{52}H_{80}O_6$ | 800.7 | violaxanthin laurate |
| 09 | 6.01 | 705.7 | 533.5 | $C_{50}H_{74}O_3$ | 722.7 | lutein caprate |
| 29 | 6.05 | 811.8 | 793.7 | $C_{54}H_{82}O_5$ | 810.8 | violaxanthin myristate |
| 27 | 6.28 | 741.6 | 653.5 | $C_{48}H_{68}O_6$ | 740.6 | violaxanthin dibutyrate |
| 10 | 6.31 | 733.5 | 533.3 | $C_{52}H_{78}O_3$ | 750.5 | lutein laurate |
| 22 | 6.52 | 821.7 | 565.5 | $C_{56}H_{86}O_5$ | 838.7 | neochrome palmitate |
| 23 | 6.55 | 707.7 | 535.6 | $C_{50}H_{74}O_2$ | 706.7 | rubixanthin caprate |
| 21 | 6.56 | 815.7 | 533.4 | $C_{58}H_{88}O_3$ | 832.7 | lutein oleate |
| 32 | 6.60 | 966.7 | 948.8 | $C_{64}H_{101}O_6$ | 965.7 | violaxanthin dilaurate |
| 11 | 6.62 | 761.8 | 533.5 | $C_{54}H_{82}O_3$ | 778.8 | lutein myristate |
| 24 | 6.81 | 735.6 | 535.4 | $C_{52}H_{78}O_2$ | 734.6 | rubixanthin laurate |
| 34 | 6.81 | 993.8 | 975.7 | $C_{66}H_{104}O_6$ | 992.8 | violaxanthin-myristate-laurate |
| 33 | 6.85 | 965.7 | 947.8 | $C_{64}H_{100}O_6$ | 964.7 | violaxanthin-myristate-caprate |
| 41 | 6.92 | 807.8 | 551.5 | $C_{56}H_{86}O_3$ | 806.8 | zeaxanthin palmitate |
| 12 | 6.94 | 789.8 | 533.5 | $C_{56}H_{86}O_3$ | 806.8 | lutein palmitate |
| 35 | 6.99 | 1021.8 | 793.7 | $C_{68}H_{108}O_6$ | 1020.8 | violaxanthin dimyristate |
| 51 | 6.99 | 735.8 | 535.5 | $C_{52}H_{78}O_2$ | 734.8 | β-cryptoxanthin laurate |
| 14 | 7.01 | 749.6 | 549.5 | $C_{64}H_{100}O_5$ | 948.6 | 5,6epoxy-luttein dilaurate |
| 13 | 7.04 | 817.8 | 533.5 | $C_{58}H_{90}O_3$ | 834.8 | lutein stearate |
| 39 | 7.04 | 1129.9 | 829.8 | $C_{76}H_{120}O_6$ | 1128.9 | violaxanthin dioleate |
| 38 | 7.11 | 1075.9 | 847.7 | $C_{72}H_{114}O_6$ | 1074.9 | violaxanthin-myristate-oleate |
| 15 | 7.17 | 733.5 | 533.3 | $C_{64}H_{101}O_4$ | 933.5 | lutein dilaurate |

*(Continued)*

**Table 2.** (Continued)

| No.[a] | $t_R$[b] | Q1(Da) | Q3 (Da) | Molecular formula | Molecular Mass | Identification |
|---|---|---|---|---|---|---|
| 20 | 7.30 | 815.7 | 533.4 | $C_{76}H_{120}O_4$ | 1096.7 | lutein dioleate |
| 53 | 7.30 | 791.9 | 535.5 | $C_{56}H_{86}O_2$ | 790.9 | β-cryptoxanthin palmitate |
| 26 | 7.31 | 791.7 | 535.4 | $C_{56}H_{86}O_2$ | 790.7 | rubixanthin palmitate |
| 17 | 7.34 | 761.8 | 533.5 | $C_{68}H_{108}O_4$ | 988.8 | lutein dimyristate |
| 43 | 7.34 | 933.9 | 533.2 | $C_{64}H_{100}O_4$ | 932.9 | zeaxanthin dilaurate |
| 45 | 7.51 | 990 | 761.8 | $C_{68}H_{108}O_4$ | 989 | zeaxanthin dimyristate |
| 18 | 7.52 | 789.8 | 533.5 | $C_{72}H_{116}O_4$ | 1044.8 | lutein dipalmitate |
| 46 | 7.63 | 989.9 | 533.4 | $C_{68}H_{108}O_4$ | 988.9 | zeaxanthin-laurate-palmitate |
| 50 | 7.73 | 1071.9 | 789.8 | $C_{74}H_{118}O_4$ | 1070.9 | zeaxanthin-oleate-palmitate |
| 47 | 7.80 | 1018.1 | 533.6 | $C_{70}H_{112}O_4$ | 1017.1 | zeaxanthin-myristate-palmitate |
| 48 | 7.92 | 789.5 | 533.5 | $C_{72}H_{116}O_4$ | 1045.1 | zeaxanthin dipalmitate |
| 49 | 8.08 | 1074.1 | 789.8 | $C_{74}H_{120}O_4$ | 1073.1 | zeaxanthin-palmitate-stearate |

carotenoids (Fig 4a). According to the "module-character" correlation analysis, the "grey" module exhibited a significant positive correlation with γ−carotene, with a correlation coefficients of 0.8. The "MEsteelblue" module was significantly positively correlated with lutein, with a correlation coefficient of 0.88.

Additionally, Gene Ontology (GO) enrichment analysis was performed for the "grey" module (Fig 4b). The results indicated that the genes in the "grey" module were associated with terms related to carotenoid synthesis and metabolism, whereas genes in the integrated module were not associated with pathways related to carotenoid metabolic process or carotenoid biosynthetic processes (Fig 4b). These findings were further analyzed.

To investigate gene expression patterns along the MEP pathway, we extracted 20 genes associated with this pathway from our expression data (Fig 4c). We found that, with the exception of the LCYB gene, all other genes had undergone duplication. Notably, the genes CHYB2, CYP97C1, and CYP97A3 each had three copies. These duplicated genes exhibited significant differences in their expression patterns across various developmental stages. For example, the expression of the LCYE-2 gene was consistently lower at each developmental stage compared to the LCYE-1 gene. These genes displayed considerable variation in expression levels across different developmental stages in the two varieties MH1 and HJ, with MH1 generally exhibiting higher expression levels than HJ. In pathway 1, the genes involved showed similar expression patterns, and their metabolites γ-carotene and zeaxanthin, exhibited comparable levels across the six developmental time points, with MH1 showing slightly higher levels than HJ at most periods (Fig 4d). In contrast, in pathway 2, the three copies of CYP97C1 and CYP97A3 exhibited significantly higher expression levels in MH1 compared to HJ at each time point. This difference was reflected in the metabolite levels, with α-cryptoxanthin and lutein being predominantly higher in MH1 than in HJ during most developmental periods (Fig 6d).

### 3.3. Antiapoptotic effects of chrysanthemum extract on ocular cells

Under the experimental conditions, chrysanthemum extract exhibited preventive effects apoptosis in zebrafish eye cells. Chrysanthemum extract concentration were set at 15.6, 31.2, and 62.5 µg/mL (n = 10). The results are presented as the fluorescence intensity of ocular apoptotic cells, as depicted in Fig 5. Detailed data are available in Supporting Information S2 Table. Medium (31.2 µg/mL) and high (62.5 µg/mL) concentrations of the extract mitigated apoptosis in ocular cells compared to the model group. Notably, the high concentration demonstrated significant effects ($p < 0.001$), surpassing those of the positive control (GSH). Previous literature has highlighted the significance of xanthophyll and zeaxanthin as essential components of macular pigments in the human cornea [10,11], capable of shielding the retina from blue light-induced damage and enhancing visual sensitivity [12]. Moreover, they have been shown to possess protective properties

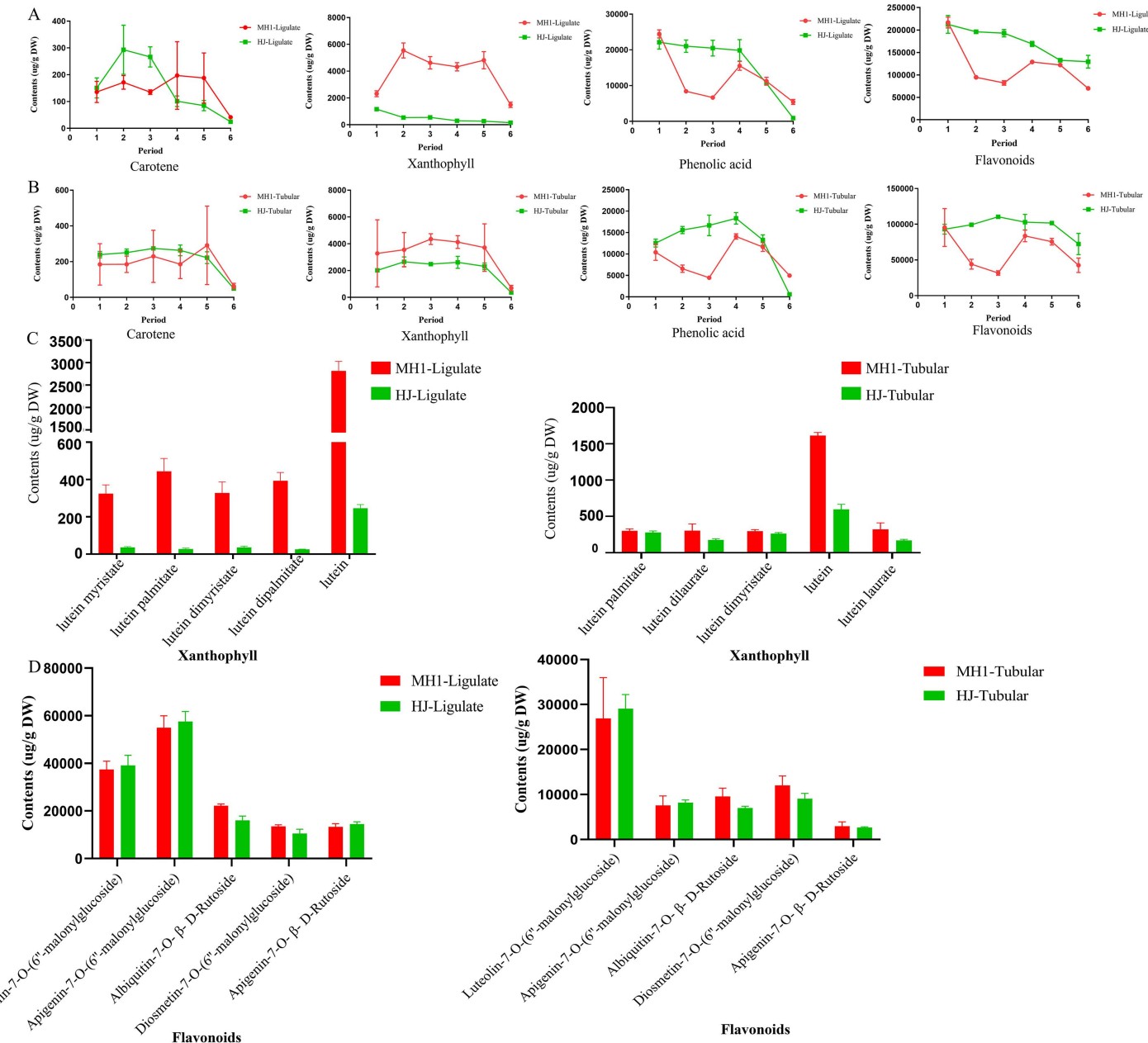

**Fig 2. Dynamic changes of carotene, xanthophyll, phenolic acid, flavonoids and top five individual components in MH1 and HJ at six different developmental stages with two flower tissues. (A)** Contents of carotene, xanthophyll, phenolic acid, and flavonoids dynamic trend in the ligulate chrysanthemum flower; **(B)** Contents of carotene, xanthophyll, phenolic acid, and flavonoids dynamic trend in the tubular chrysanthemum flower; **(C)** Contents of the top five xanthophll in ligulate and tubular flower of MH1 and HJ at the highest mature stage; **(D)** Contents of top five flavonoids in ligulate and tubular flower of MH1 and HJ at the highest mature stage.

against the development of age-related macular degeneration in numerous studies [11,13]. Consequently, lutein is frequently utilized to prevent and decelerate the progression of degenerative eye conditions such as cataracts, age-related macular degeneration, and retinal nerve diseases [14], underscoring the beneficial activities of the abundant carotenoids identified in the new chrysanthemum MH1 cultivar.

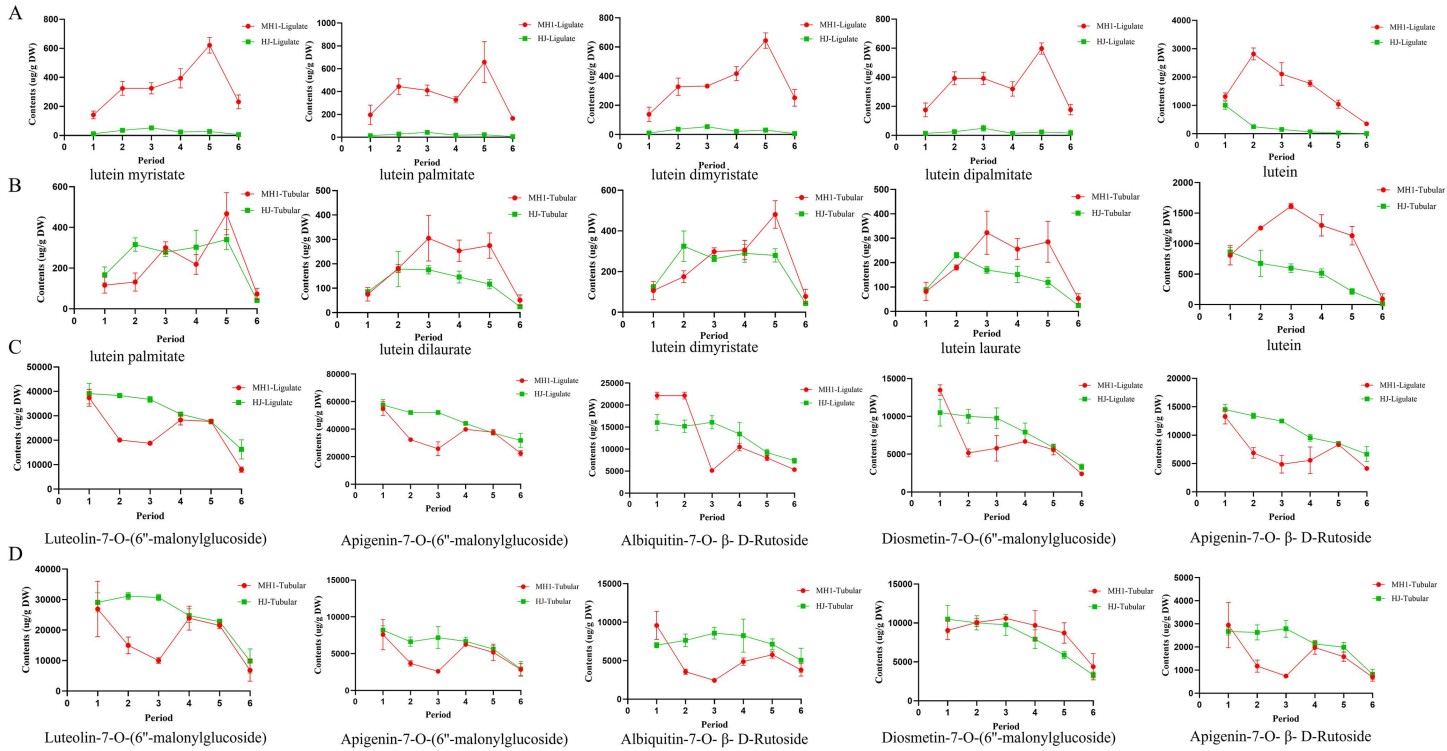

**Fig 3. Contents dynamic trend of top five individual components of xanthophyll and flavonoids in ligulate chrysanthemum flower and tubular chrysanthemum flower. (A)** Xanthophyll in ligulate chrysanthemum flower; **(B)** Xanthophyll in tubular chrysanthemum flower; **(C)** Flavonoids in ligulate chrysanthemum flower; **(D)** Flavonoids in tubular chrysanthemum flower.

### 3.4. Efficacy of chrysanthemum extract in preventing retinopathy

We assessed the ocular choroidal vascular area of zebrafish in the control (n = 10), treatment (15.6, 31.2, 62.5 µg/mL, n = 10), and Augentropfen Stulln® Mono groups (n = 10). Comparative analysis revealed that the ocular choroidal vascular area was significantly larger in the model group compared to the control group ($p < 0.01$). This observation suggests that $CoCl_2$ induced tissue hypoxia, thereby promoting choroid neovascularization in adult zebrafish, and the establishment of a wet macular degeneration model was successful. In comparison to the model group, the medium (31.2 µg/mL) and high (62.5 µg/mL) concentrations of the extract exhibited a decrease in the vessel area in zebrafish, albeit not statistically significant. These results illustrate that chrysanthemum extract reduces retinal choroid neovascularization in zebrafish. Fig 6 illustrates that the distinct trend of different concentrations of the chrysanthemum extract in prevent retinopathy.

The eye-protective effects of chrysanthemum extract align with findings from previous studies. Wu et al. demonstrated that wild chrysanthemum eye drops effectively mitigate human dry eye syndrome [15], offering improvements in cases of mild and moderate dry eye syndrome. Gong et al. (2002) [16] reported that chrysanthemum water extract significantly enhances electrophysiological function on ERG in rats, as along with improving retinal morphology as observed through staining with optimal cutting temperature compound and hematoxylin–eosin. This beneficial effect may be attributed to the presence of carotenoids in chrysanthemum water extracts [17]. Moreover, ocular carotenoids act as natural fat-soluble antioxidants, capable of scavenging oxygen free radicals induced by light exposure [18–20], thereby safeguarding the eyes against oxidative stress, cell apoptosis, mitochondrial dysfunction, and inflammation [21–24].

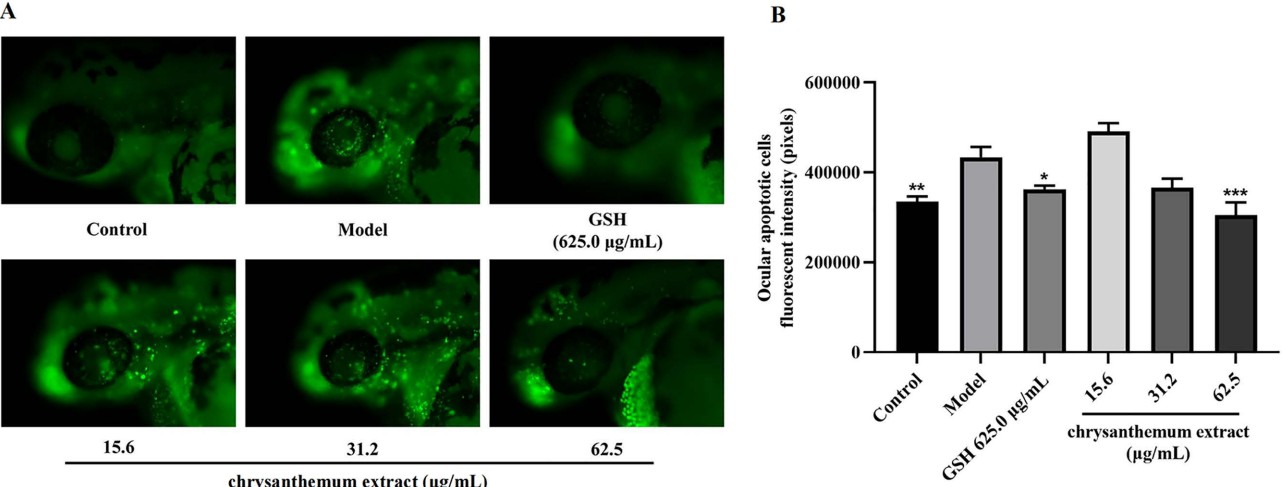

**Fig 4. Weighted gene coexpression network analysis (WGCNA).** (a) distribution of 85912 genes in the modules and the correlation between each module and α-carotene, α-cryptoxanthin, zeaxanthin, lutein and γ−carotene. **(b)** GO enrichment results of "Grey" module. **(c)** Pathways related to carotenoid metabolism in chrysanthemum petals. **(d)** Measurement results of five metabolites' concentrations in chrysanthemum petals across six different growth stages.

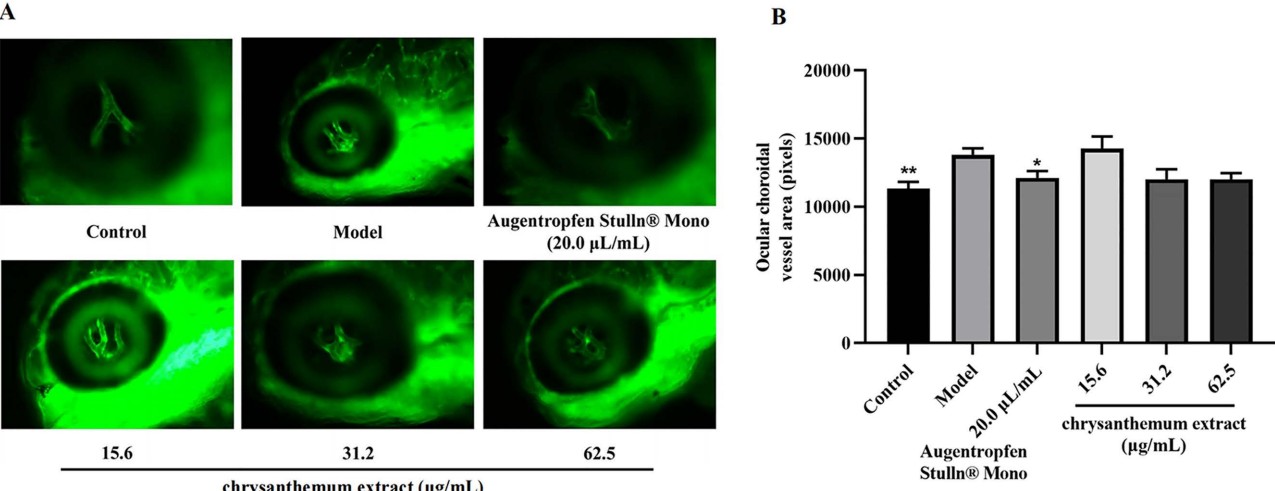

**Fig 5. Fluorescence intensity of zebrafish eye apoptotic cells treated with chrysanthemum extract compared with the model control group, *p<0.05, **p<0.01, ***p<0.001.**

## 4. Discussion and conclusions

This study identifies xanthophyll-enriched carotenoids-rather than flavonoids-as the principal contributors to the ocular protective effects of chrysanthemum, with the newly developed yellow-petaled cultivar MH1 showing a particularly high abundance of lutein-type compounds and efficacy in zebrafish models. These findings refine the prevailing view that chrysanthemum's eye-health benefits arise mainly from flavonoids (e.g., quercetin and luteolin), and instead position carotenoid composition-especially xanthophyll load and profile-as a primary determinant of bioactivity..

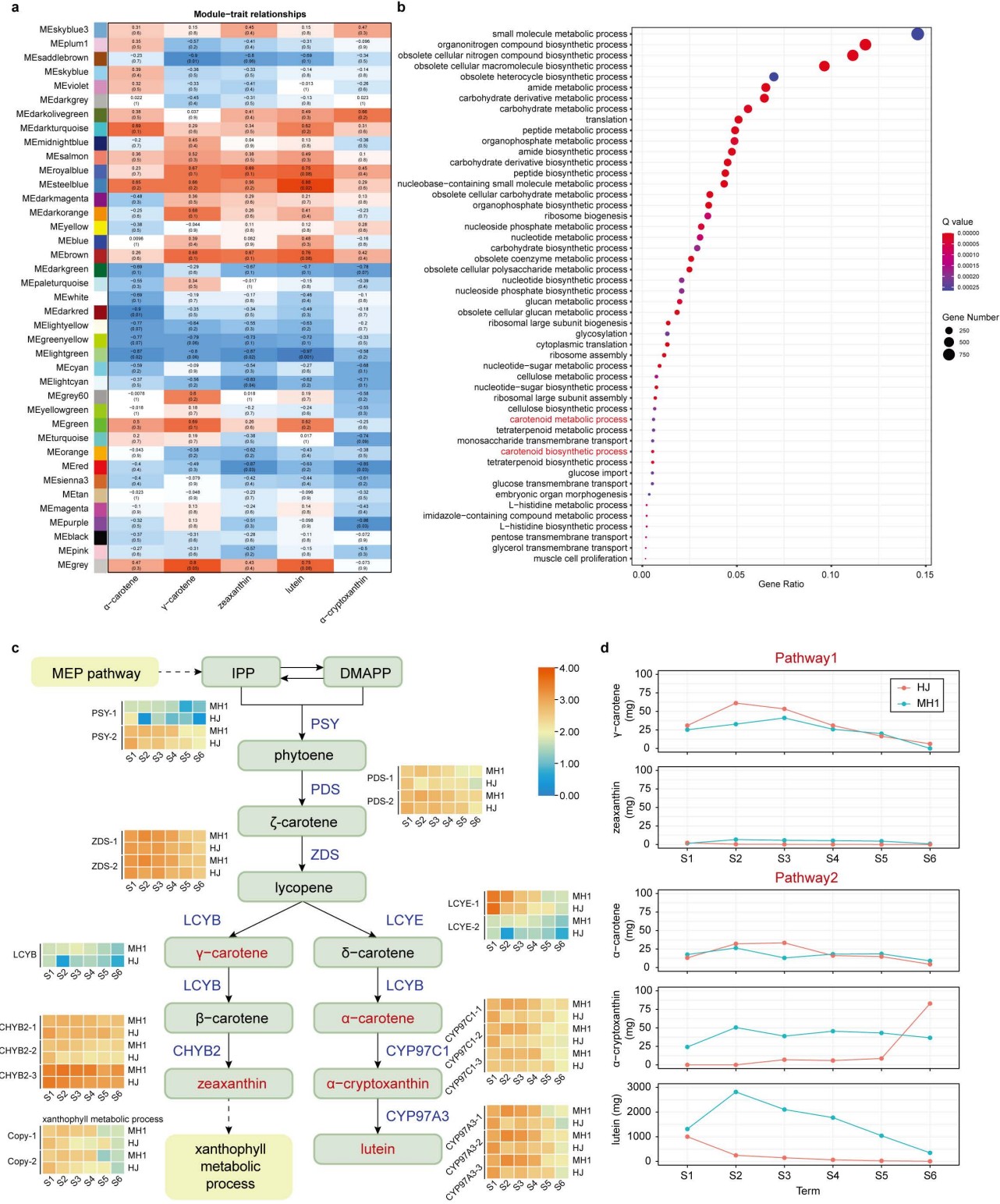

**Fig 6. Eye choroidal vascular area of zebrafish treated with chrysanthemum extract compared with the model control group, *p<0.05, **p<0.01.**

### 4.1. Mechanistic considerations

Xanthophylls such as lutein and zeaxanthin possess dual functionality relevant to retinal protection: (i) physical blue-light filtering and (ii) chemical quenching of reactive oxygen species (ROS). In the zebrafish assays used here—where photic injury and oxidative stress play central pathogenic roles—the robust activity of MH1 extracts is consistent with an anti-oxidant and photoprotective mode of action. Beyond direct radical scavenging, xanthophylls may engage endogenous cytoprotective pathways (e.g., Nrf2-HO-1 axis), dampen NF-κB–mediated inflammatory signaling, stabilize mitochondrial function in photoreceptors, and support membrane integrity due to their orientation within lipid bilayers. Although flavonoids also have antioxidant and anti-inflammatory properties, our composition–efficacy correlation across two varieties indicates that the variance in ocular endpoints tracks more closely with xanthophyll content than with total flavonoids. This does not exclude synergism; indeed, co-occurrence of flavonoids could regenerate carotenoid radicals or modulate absorption and tissue distribution. Disentangling additivity versus synergy will require fractionation, recombination, and isobologram analyses.

### 4.2. Coloration biology as a phenotypic proxy for bioactivity

The tight correlation between yellow petal coloration and xanthophyll abundance provides a convenient phenotypic handle to enrich for ocular bioactivity in breeding programs. In chrysanthemums, anthocyanins dominate red/pink palettes, whereas carotenoids—particularly lutein-type xanthophylls—underlie yellow hues. Our data reinforce that visual phenotype can be used as an initial surrogate marker for a favorable metabolite profile, though targeted quantification remains essential because petal chroma can be modulated by plastid development, carotenoid esterification, and carotenoid cleavage.

### 4.3. Pathway insights and targets for metabolic engineering

Transcriptome-guided network analysis in MH1 versus comparators could pinpoint rate-limiting steps and regulatory nodes. Practically, breeding or editing strategies that (1) increase upstream flux, (2) favor α-branch allocation (higher LCYE/LCYB ratio), (3) enhance hydroxylation efficiency, and (4) temper CCD-mediated cleavage should raise lutein pools while preserving floral fitness. Plastid biogenesis genes and esterification capacity also merit attention, as ester forms improve storage stability in petals and may influence extract stability.

### 4.4. Comparison with prior chrysanthemum literature

Historically, eye-protective claims for chrysanthemum have emphasized flavonoids, often inferred from in vitro antioxidant assays. Our results suggest that when endpoints reflect photoreceptor integrity and in vivo oxidative stress (as in zebrafish), xanthophyll variation explains efficacy more directly than total phenolics. This shifts the emphasis from generic antioxidant capacity toward pigment-specific, retina-relevant chemistry. The finding aligns with the recognized role of lutein/zeaxanthin in vertebrate retinal tissues and supports the rationale for prioritizing xanthophyll metrics in evaluating ocular nutraceutical candidates from ornamentals.

### 4.5. Formulation, stability, and bioavailability

The translational value of MH1 extracts will hinge on xanthophyll chemistry and matrix effects. Lutein in chrysanthemum often occurs as esters, which can enhance stability during processing yet require hydrolysis for absorption. Co-extracted lipids and the choice of delivery system (oil suspensions, emulsions) will affect bio-accessibility. Light/heat/oxygen can isomerize or degrade carotenoids; standardized processing that limits isomerization and preserves all-trans lutein may improve potency and batch-to-batch consistency. Because flavonoids may influence intestinal transporters (e.g., SR-B1) or micellarization, preserving a native co-matrix could be advantageous despite xanthophyll dominance in efficacy attribution.

### 4.6. Implications for breeding and product development

Color is already a key commercial trait in chrysanthemums; our data add a functional dimension by linking yellow chroma to ocular health potential. This creates a dual-objective breeding paradigm: ornamental value plus validated bioactivity. Marker-assisted selection using expression or SNP markers for LCYE/LCYB/CHY genes, combined with rapid pigment profiling (HPLC-DAD or LC-MS), could accelerate development of xanthophyll-rich lines. From a product standpoint, MH1 provides a traceable cultivar source for standardized extracts, enabling tighter control over active content, regulatory documentation, and clinical translation.

### 4.7. Limitations and future directions

First, the causal primacy of xanthophylls is supported by correlation and pharmacology but not yet proven by gain-/loss-of-function in extracts. Bioactivity-guided fractionation, carotenoid depletion/reconstitution, and testing of purified lutein vs. matched flavonoid fractions will be decisive. Second, zebrafish offer strong throughput and retinal homology but are not a substitute for mammalian validation; mouse light-damage and genetic retinal degeneration models should be pursued. Third, we did not map the full stereochemistry and ester distribution of lutein in MH1; these features can modulate absorption and tissue targeting. Fourth, the possibility of minor yet potent constituents (e.g., apocarotenoids) contributing to efficacy warrants untargeted metabolomics.

## Supporting information

**S1 Fig. The change trend of compound contents and expression amount trend of the related genes.**
(TIF)

**S1 Table. Raw data of carotenoid content determination.**
(XLS)

**S2 Table. Experimental results of chrysanthemum extract eye protection effect.**
(XLSX)

## Author contributions

**Conceptualization:** Jing Zhang.

**Data curation:** Gangqiang Dong.

**Formal analysis:** Yifan Wang.

**Funding acquisition:** Gangqiang Dong, Sha Chen.

**Investigation:** Dan Yang, Xaiofei Liu, Zhuyin Chen, Yanyan Su, Yan Liu, Jingjing Zhu.

**Methodology:** Gangqiang Dong.

**Project administration:** Jing Zhang.

**Resources:** Sha Chen.

**Supervision:** Yan Liu, Jingjing Zhu, Jun Zhang, Jing Zhang, Sha Chen.

**Validation:** Dan Yang, Xaiofei Liu, Zhuyin Chen, Yanyan Su, Jun Zhang.

**Visualization:** Yifan Wang, Hedi Zhao.

**Writing – original draft:** Yifan Wang, Hedi Zhao, Sha Chen.

**Writing – review & editing:** Sha Chen.

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
