## [Decision Letter · Decision Letter 0]

9 Oct 2025

Dear Dr. Chen,

Thank you for submitting your manuscript to PLOS ONE. After careful consideration, we feel that it has merit but does not fully meet PLOS ONE’s publication criteria as it currently stands. Therefore, we invite you to submit a revised version of the manuscript that addresses the points raised during the review process.

We look forward to receiving your revised manuscript.

Kind regards,

Fahrul Nurkolis

Academic Editor

PLOS ONE

Journal Requirements:

2. To comply with PLOS One submissions requirements, in your Methods section, please provide additional information regarding the experiments involving animals and ensure you have included details on (1) methods of sacrifice, (2) methods of anesthesia and/or analgesia, and (3) efforts to alleviate suffering.

5. We note that your Data Availability Statement is currently as follows: All relevant data are within the manuscript and in Supporting Information files.

Reviewers' comments:

Reviewer's Responses to Questions

**Comments to the Author**

1. Is the manuscript technically sound, and do the data support the conclusions?

Reviewer #1: Yes

Reviewer #2: Partly

2. Has the statistical analysis been performed appropriately and rigorously?

Reviewer #1: Yes

Reviewer #2: Yes

3. Have the authors made all data underlying the findings in their manuscript fully available?

Reviewer #1: Yes

Reviewer #2: No

4. Is the manuscript presented in an intelligible fashion and written in standard English?

Reviewer #1: Yes

Reviewer #2: Yes

Reviewer #1: The manuscript 'Integrated Metabolomics and Bioactivity Analysis of new Chrysanthemum cultivar Petals: Insights into Eye-Protecting Agents' reports a very thorough, well-conducted analysis of the molecular content and synthesis in Chrysanthemum flowers. As such, it is no doubt that this part is worth publishing, eventually in its present form.

Not being an expert in assessing the bioactivity of molecular compounds, I cannot judge the part of the paper concerning the ests in Zebrafish. However, again, the metabolomic and transcrptomic part of this paper is certianly valuable enough to be published.

Reviewer #2: Comments on the article- PONE-D-25-34406ss

Line 21 (Abstract): should be ‘Economical’ and not ‘Economically’

Line 38, 39 (Introduction): Check the grammar

Line 92 (Materials n Methods): check the sentence

Line 127 (section 2.2.1): Check the sentence

Line 160-162 (section 2.4): Please specify if there are two mobile phases and hence rectify the sentence

Section 2.9 (Statistical analysis): The statistical tests appear to be applied appropriately. However, the rationale or the specific purpose behind each test could be clarified better; otherwise, readers less familiar with metabolomics would find this, a bit difficult to comprehend.

Table 1: It is unclear as to why the compound ‘Acacetin-7-O-β-D-rutinoside’ is highlighted out of all other compounds in table 1. Mentioning the reason in the table caption or in a relevant section would give the table better clarity.

Results- The results are presented in detail, but in several cases their broader biological or functional implications are not clearly explained. A more explicit interpretation of what these findings mean in the context of the study objectives would greatly benefit readers. In short a ‘Discussion’ or ‘Interpretation’ section would further strengthen the manuscript.

Overall comment- The manuscript presents a well-structured study with detailed experimental work and clear presentation of results. The use of advanced analytical and statistical methods is appropriate and supports the reliability of the findings. However, the rationale for certain analyses and the broader implications of the results are not always explained in sufficient detail, which may make the manuscript less accessible to readers outside the immediate field. Providing clearer interpretation and including a dedicated discussion section would enhance the clarity, impact, and overall readability of the study.

**Do you want your identity to be public for this peer review?** For information about this choice, including consent withdrawal, please see our Privacy Policy

Reviewer #1: No

Reviewer #2: No

---

## [Author Response · Author response to Decision Letter 1]

2 Dec 2025

Thank you for your email and for bringing the change in the author list to our attention. We sincerely apologize for the oversight and any inconvenience it may have caused. The alteration was unintentional and resulted from an operational error while we were updating the submission system. During the process of designating Dr. Gangqiang Dong as a co-corresponding author- a role we intended to add to acknowledge his pivotal contributions to this new variety-we inadvertently deleted the entire author list and had to re-enter it manually. We would like to assure you that the sequence of authors has not been altered in any way.

We have fully addressed all reviewer and editor comments: 1) Ensured the manuscript complies with PLOS ONE’s style and file naming requirements; 2) Supplemented animal experiment details (sacrifice method, anesthesia/analgesia, and suffering-alleviation efforts); 3) Removed all funding-related text from the manuscript and updated the ‘Funding Information’ section with correct grant numbers; 4) Included all raw data necessary for replicating the study results.

---

## [Decision Letter · Decision Letter 1]

16 Dec 2025

Integrated Metabolomics and Bioactivity Analysis of new Chrysanthemum cultivar Petals: Insights into Eye-Protecting Agents

PONE-D-25-34406R1

Dear Dr. Chen,

We’re pleased to inform you that your manuscript has been judged scientifically suitable for publication and will be formally accepted for publication once it meets all outstanding technical requirements.

Kind regards,

Fahrul Nurkolis

Academic Editor

PLOS One

Additional Editor Comments (optional):

Accept!

Reviewers' comments:

Reviewer's Responses to Questions

**Comments to the Author**

Reviewer #1: All comments have been addressed

2. Is the manuscript technically sound, and do the data support the conclusions?

Reviewer #1: Yes

3. Has the statistical analysis been performed appropriately and rigorously?

Reviewer #1: Yes

4. Have the authors made all data underlying the findings in their manuscript fully available?

Reviewer #1: Yes

5. Is the manuscript presented in an intelligible fashion and written in standard English?

Reviewer #1: Yes

Reviewer #1: The authors have answered my main comments. I recommend that this paper be accepted in its present form.

**Do you want your identity to be public for this peer review?** For information about this choice, including consent withdrawal, please see our Privacy Policy

Reviewer #1: No

---

## [Editor Report · Acceptance letter]

PONE-D-25-34406R1

PLOS One

Dear Dr. Chen,

I'm pleased to inform you that your manuscript has been deemed suitable for publication in PLOS One. Congratulations! Your manuscript is now being handed over to our production team.

Kind regards,

on behalf of

Dr. Fahrul Nurkolis

Academic Editor

PLOS One